# Applicability of Radiomics for Differentiation of Pancreatic Adenocarcinoma from Healthy Tissue of Pancreas by Using Magnetic Resonance Imaging and Machine Learning

**DOI:** 10.3390/cancers17071119

**Published:** 2025-03-27

**Authors:** Dimitrije Sarac, Milica Badza Atanasijevic, Milica Mitrovic Jovanovic, Jelena Kovac, Ljubica Lazic, Aleksandra Jankovic, Dusan J. Saponjski, Stefan Milosevic, Katarina Stosic, Dragan Masulovic, Dejan Radenkovic, Veljko Papic, Aleksandra Djuric-Stefanovic

**Affiliations:** 1Center for Radiology, University Clinical Centre of Serbia, Pasterova No. 2, 11000 Belgrade, Serbia; 2School of Electrical Engineering, University of Belgrade, Bulevar kralja Aleksandra 73, 11120 Belgrade, Serbia; milica.badza@ic.etf.bg.ac.rs (M.B.A.);; 3Innovation Center of the School of Electrical Engineering in Belgrade, Bulevar kralja Aleksandra 73, 11120 Belgrade, Serbia; 4Department for Radiology, Faculty of Medicine, University of Belgrade, Dr Subotica No. 8, 11000 Belgrade, Serbia; 5Department for HBP Surgery, Clinic for Digestive Surgery, University Clinical Centre of Serbia, Koste Todorovica Street, No. 6, 11000 Belgrade, Serbia; 6Department for Surgery, Faculty of Medicine, University of Belgrade, Dr Subotica No. 8, 11000 Belgrade, Serbia

**Keywords:** pancreatic adenocarcinoma, magnetic resonance imaging, radiomics, machine learning

## Abstract

Pancreatic carcinomas rank as the second most common cause of death in the group of digestive tract cancers, primarily because they are clinically “silent” for a long time until their spread infiltrates some of the adjacent anatomical structures. The most common ways to detect pancreatic adenocarcinoma are by ultrasonography (US), endoscopic ultrasonography (EUS), computed tomography (CT), and magnetic resonance imaging (MRI) with magnetic resonance cholangiopancreatography (MRCP). The early detection of pancreatic carcinoma can reduce the mortality rate. Radiomics has a leading role in the machine learning classification and early detection of this disease. Using only magnetic resonance imaging and a machine learning approach for the diagnosis of pancreatic adenocarcinoma based on radiomics may speed up the diagnostic process, as well as enable preventive diagnostics.

## 1. Introduction

Pancreatic adenocarcinoma is in the second place as the cause of death in the group of cancers of the digestive tract, right after colorectal cancer, with a constant increase in incidence [1]. The ductal type of adenocarcinoma accounts for more than 90% of pancreatic cancers [2]. Pancreatic carcinomas remain clinically “silent” for a long time until their spread infiltrates some of the neighboring anatomical structures, when the tumor is usually already 2–5 cm in size, unresectable or borderline resectable, and locoregionally or distantly disseminated; this is the main cause of the high mortality rates and very unfavorable long-term prognosis of this malignant tumor [2]. About 90% of patients with pancreatic adenocarcinoma have an elevated serum concentration of carbohydrate antigen (CA 19-9), which is currently the only widely used tumor marker for adenocarcinomas of biliopancreatic origin [2].

It is not always easy to distinguish normal pancreatic tissue from adenocarcinoma. The diagnostic algorithm for pancreatic cancer using radiological methods begins with ultrasound, the possibilities of which are partially limited due to the presence of gas in the digestive tube, followed by computerized tomography (CT), which makes it impossible to detect tumors smaller than 2 cm in almost 10% of cases, endoscopic ultrasonography (EUS), or magnetic resonance imaging (MRI) which, together with magnetic resonance cholangiopancreatography (MRCP), can be used to detect and characterize tumors smaller than 3 cm [3,4,5]. Radiomics can help identify subtle patterns in imaging data that indicate the presence of pancreatic tumors, even at early stages when they may be difficult to detect using conventional methods. This could potentially lead to earlier diagnosis, when treatment options are more effective.

Despite its continuous increase in incidence and high mortality, pancreatic cancer has long been considered unsuitable for screening programs [6]. However, due to the severity of the disease and consequently great need for early diagnosis, the use of precision medicine methods, among which radiomics has a leading role, is a leading trend in medicine [7,8,9]. The majority of studies have used radiomics data from CT examinations and only a few studies have used MRI data, despite its superior tissue contrast capabilities [7,8,9]. Thus, the objective of our study was to assess the applicability of radiomics extracted from selected MR images for the differentiation of pancreatic adenocarcinoma from surrounding healthy pancreatic tissue by using machine learning (ML). However, the first aim was to analyze the accuracy of different ML classifier models and different radiomics feature reduction methods in differentiating pancreatic adenocarcinoma from normal pancreatic tissue on selected MR images. The second aim is to possibly create a shortened MRI protocol for screening purposes using only two non-contrast sequences.

## 2. Materials and Methods

### 2.1. Patients

One hundred forty-five patients with histologically proven pancreatic adenocarcinoma who underwent a MR examination during the period from 2013 to 2024 in the Department of Digestive radiology, Center for Radiology, University Clinical Center of Serbia were included in this study. The Institutional Ethics Board approved this retrospective study and informed consent was waived. The criteria for inclusion were (1) the histological confirmation of pancreatic adenocarcinoma obtained by biopsy or after surgery and (2) an abdominal MR exam performed in our institution according to the following protocol. The exclusion criteria were (1) prior chemotherapy and/or radiation therapy of pancreatic carcinoma and (2) the insufficient quality of an MRI exam for accurate interpretation.

### 2.2. MRI Scanning Protocol

All patients underwent MRI in a 1.5-T scanner (Signa HDxt, GE Healthcare, Chicago, IL, USA). An abdominal MR exam was performed using an eight-channel phased-array abdominal coil and spine array coil to optimize the signal-to-noise ratio (SNR). An axial breath-hold T2-weighted (T2W) single-shot fast spin echo sequence, a breath-hold T2W fat-suppressed (T2W-FS), and a breath-hold 3D volumetric T1W-FS GRE sequence before and after the intravenous application of paramagnetic gadolinium-based contrast agent (gadobutrol) were performed (Table 1). They were followed by a respiratory triggered single-shot echo-planar imaging DWI sequence with two diffusion sensitivity values (b 0 and 800 s/mm^2^) (Table 1). Quantitative ADC maps for the combination of b-values 0 and 800 s/mm^2^ were generated by commercial workstation software (Advantage Windows 4.3, GE Health-care Technologies, Chicago, IL, USA) using the standard mono-exponential model. At the end of the MR examination, after the native T1W-FS sequence, contrast was administered and a 3D-dynamic three-phase post-contrast study was done during the arterial, portal, and delayed (venous) phases (Table 1). This MRI protocol was followed by 3D-MRCP in the coronal plane.

### 2.3. MRI Analysis

#### 2.3.1. Database

We used two image databases; the first for the in-depth analysis of different ML classifiers and feature extraction methods and the second for the testing of the given results.

The first image database (for training and validation purposes) was gathered from the 2013 to 2019 period and consists of T2W-FS and ADC MR images of 87 patients.

The second image database (for testing purposes) was gathered from the 2020 to 2024 period from the MR examinations performed in the same department by the same MR scanner using the same scanning protocol and also consists of T2W-FS and ADC MR images of 56 patients.

#### 2.3.2. Image Processing

For each patient, three nearly consecutive slices on the largest tumor area were selected. Three slices were chosen because even the smallest tumors were detectable on three consecutive slices, so we could have a uniform image set and as many images as possible from all patients. Each tumor location was recorded and the largest tumor diameter was measured on the corresponding axial T2W-FS and ADC MRI images. The freely available software package MaZda v4.6 (Institute of Electronics, Technical University of Lodz, Poland) [10] was used for the segmentation of the pancreas and tumor tissues and feature extraction (Figure 1 and Figure 2). One radiologist with six years of experience in general radiology (first author) performed the segmentation, but in challenging situations two senior radiologists were consulted beforehand.

Since three nearly consecutive slices for pancreas and tumor were segmented for each patient, this provided 261 slices for feature extraction for tumor and 261 slices for pancreas in the first group, as well 168 slices for pancreas and 168 slices for tumor in the second group.

MaZda is capable of obtaining 304 radiomics features from six categories: First-order histogram, co-occurrence matrix, run-length matrix, absolute gradient, autoregressive model parameters, and wavelet analysis [10]. In total, in the training and validation group, 79,344 feature values were gathered for classification for each category, for T2W-FS and ADC images separately. In the testing group, MaZda was also used for feature extraction, but only selected the most relevant features observed after the analysis of the first data set.

#### 2.3.3. Machine Learning Classification

Tumor classification was performed using six different ML classifiers in Python v3.7: Logistic Regression (LR), k-Nearest Neighbors (KNN), Support Vector Machine (SVM), Neural Network (NN), Random Forest (RF), and Gaussian Naïve Bayes (GNB). The flowchart of the analysis is shown in Figure 3. The performance of each classifier was tested using 5-fold subject-wise cross-validation, i.e., data from one subject could be found either in the training or validation set. In this manner, the generalization capability for medical practice of the classifiers was tested. In order to be able to compare the results of tumor classification between T2W-FS and ADC MR images, the division of the data through folds was fixed—the same division of patients’ data into folds was used for all evaluations.

Data imputation was performed to overcome the lack of feature values after the process of segmentation and feature extraction within the MaZda package. The features that had more than 50% of missing values in the training set were removed from further analysis from both the training and validation set. The substitution of missing data was performed using the mean value of the existing feature values from the training set, in both sets. This process was performed for each training–validation set division during the cross-validation process.

Feature reduction was performed in order to obtain a ratio between the number of features and number of patients that was suitable for ML algorithm training [11]. Having that in mind, the number of features was reduced to 10% of the number of patients in the training–validation group, which resulted in 9 features for further pancreatic tumor classification [11]. A total of 6 different feature reduction methods were analyzed: one feature extraction method—Principal Component Analysis (PCA)—and 5 feature selection methods—Fisher Score, Mutual Information, the Anova test, and two Minimum Redundancy Maximum Relevance methods, one based on Mutual Information (MRMR Mutual Information) and the other on the Anova test (MRMR Anova).

During the training process, the optimization of classifiers’ hyperparameters through grid search with 5-fold cross-validation was performed. The used classifiers with their observed hyperparameters are listed in Table 2; the parameters that are not listed were used with their default values.

The in-depth analysis of different ML classifiers and different feature reduction methods was performed on the first image database and the results were tested on the second image database.

Tumor classification and feature reduction was performed in Python v3.7. The performance of classifiers was evaluated using the classification metrics of accuracy (Acc), sensitivity (Se), specificity (Sp), F1-score, and area under the curve (AUC).

### 2.4. Statistical Analysis

Pearson’s chi-square test or Fisher’s exact probability test was used to assess differences in clinical categorial variables (gender of patients, location of tumor, and frequency of tumors smaller than 2 cm) between the training–validation and test group. After testing the normality of distribution of the numerical data (age of patients and diameter of tumor), the Mann–Whitney test was used to assess differences between the training–validation and test group. A *p* value less than 0.05 was considered statistically significant. Statistical analysis of clinical data was performed by using SPSS software (SPSS for Windows; Version 26.0, Chicago, IL, USA).

## 3. Results

The demographic characteristics of patients and characteristics of pancreatic tumor in the training–validation and test group are presented in Table 3.

There were no significant differences in the age and gender of patients nor in tumor location between the training–validation and test cohort, but the average size of tumor was significantly smaller in the test group (*p* = 0.001), as was the percentage of tumors smaller than 2 cm (*p* = 0.009).

### 3.1. Machine Learning Classification Results

The tumor classification was performed using six different classifiers with six different feature reduction methods through five-fold cross-validation on the first image database. The distribution of Acc for all feature reduction methods for both T2W-FS and ADC MR images is shown in Figure 4A,B. The distribution is shown only for the best classification model for that specific reduction method.

It could be observed, for both T2W-FS and ADC MR images, that the usage of a Mutual Information score for feature selection with an RF classifier shows the best performance in terms of the mean value of Acc for five-fold cross-validation, as well as the dispersion of Acc values, which could be attributed to the stability of the classifier across folds (Figure 4A,B).

### 3.2. Feature Reduction

The feature reduction was performed for each training–validation data division separately. The feature occurrence through cross-validation for each feature reduction method is shown in Figure 5 and Figure 6 for T2W-FS and ADC MR images, respectively.

It could be observed that the most valuable information for tumor classification could be found in the co-occurrence matrix group of features (Figure 5 and Figure 6). This is not unexpected, due to the fact that this group contains the most features, at 42 features of the total 61 (features that remain after data imputation). Further, the most valuable features are related to the area of the regions of interest (pancreatic tumor and healthy tissue) and border pixels and their correlation, as well as their intensity itself. Thus, the developed model can be considered naturally more intuitive to understand and less of a black box input–output approach.

The feature reduction method based on the Mutual Information score has the best stability for T2W-FS MR images, where for all five folds, seven out of nine selected features were the same (Figure 5). For ADC MR images, the best stability for feature reduction was based on the Anova and Fisher’s score, with seven out of nine selected features being the same (Figure 6). The feature reduction based on Mutual Information gave the same five out of nine selected features for ADC MR images (Figure 6).

In order to choose the best performing classifier–feature-reduction combination, the stability of the trained model through cross-validation was observed. For both T2W-FS and ADC MR images, the RF classifier with the feature reduction based on the Mutual Information score showed the highest mean value of Acc. All the metrics for the fold with the best results for the RF classifier in combination with Mutual Information are shown in Table 4 and Table 5 for T2W-FS and Table 6 and Table 7 for ADC MR images. In Table 4 and Table 6, the results for all classifiers with feature reduction based on the Mutual Information score are presented for T2W-FS and ADC MR images. In Table 5 and Table 7, the results for the RF classifier with all feature reduction methods are shown for T2W-FS and ADC MR images.

It could be observed that for pancreatic tumor classification based on features, better results were achieved with ADC MR images. Even though the KNN classifier with feature reduction based on Mutual Information for ADC MR images has a slightly better performance than the RF classifier, it is shown that RF has better stability throughout the folds, with a mean value and standard deviation of Acc of 93.37 ± 3.58% for RF and 91.61 ± 4.95% for KNN (Table 6).

### 3.3. Classification Model of Pancreatic Adenocarcinoma

Based on the previously shown results, as the final classification model for pancreatic tumor classification, the RF classifier with feature reduction based on the Mutual Information score was chosen. The hyperparameters that were chosen through grid-search with five-fold cross-validation for the RF classifier, as well as the chosen nine features after the feature reduction based on the Mutual Information score, are shown in Table 8 for both T2W-FS and ADC images.

It can be observed that the only difference between the two RF models’ hyperparameters is the Criterion. The selected features for both T2W-FS and ADC MR images are all from the same category: the co-occurrence matrix.

### 3.4. Testing of Classification Model

After the in-depth analysis, the obtained outcomes (nine selected features and the RF classifier with selected hyperparameters) were used and tumor classification was performed and tested on the second image database. The given results are shown in Table 9 for T2W-FS and ADC MR images.

## 4. Discussion

In our study, we compared six different feature reduction methods and six different ML classifiers on the radiomics features extracted from the T2W-FS and ADC MR images of patients with pancreatic adenocarcinoma through five-fold cross-validation in order to differentiate adenocarcinoma from normal pancreatic tissue. The combination of the Mutual Information score for feature selection with an RF classifier showed the highest accuracy (AUC 0.94 and 0.98) and stability for both T2W-FS and ADC MR images, respectively. For pancreatic tumor classification based on radiomic features, better results were achieved with ADC MR images. All nine selected features for both T2W-FS and ADC MR images are from the same category: the co-occurrence matrix. The prediction model constructed of the nine most relevant radiomics features achieved moderate diagnostic performance in the test group (AUC 0.69 and 0.81 for T2W-FS and ADC MR images).

The data imputation technique that was implemented was a widely used, simple statistical method—mean value, which tends to give some unrealistic data. In this manner, we also set the cut-off criteria for removing the features that had more than 50% of data missing, which will only lead to more unrealistic data. Based on the results obtained on the test and validation sets, it can be concluded that the data imputation technique influenced the further analysis of the classification and feature selection methods in such a way that it made a greater difference in the accuracy of the classification.

To the best of our knowledge, until now there has been a lack of studies in the literature on using MRI radiomics data for the differentiation of pancreatic adenocarcinoma from normal pancreatic tissue [7,8,9]. Thus, a direct comparison of our results was not possible. However, two studies accessed the accuracy of radiomics analysis extracted from CT imaging data in the differentiation of pancreatic ductal adenocarcinoma (PDAC) from normal pancreas tissue [12,13]. Superior accuracy was reported in both, with an AUC of 0.99 in the study of Chu and Park et al. [12] and 0.97/0.93 in the study of Chen and Chang et al. for local/external testing [13]. In both studies, the freely available PyRadiomics (version 2.2.0) software was used for radiomics feature extraction [12,13]. In the study of Chu and Park et al., the whole 3D volume segmentation of pancreas was conducted in the venous phase of the CT examinations of 190 patients with PDAC (mean size of 4.1 ± 1.7 cm) and 190 healthy control cases [12]. The forty most relevant radiomics features were selected for the final classification of the PDAC of 478 available features by using the MRMR based on Mutual Information selection method and an RF classifier, and the model achieved excellent performance in differentiating PDAC cases from control cases with normal pancreas, with a reported sensitivity of 100%, specificity of 98.5%, and accuracy of 99.2% [12]. When their prediction model constructed of the five most relevant radiomics features was assessed, it achieved a sensitivity of 95%, specificity of 92.3%, and accuracy of 93.6% in the validation set [12], which is close to our results in the training–validation group, but much better than our results in the test group.

In the large-cohort study of Chen and Chang et al., two methods of segmentation were performed on the portal venous phase CT images: patient-based segmentation of whole pancreas together with the tumor, like in the previous study [12], and patch-based segmentation of tumor ROIs within a pancreas [13], similarly as we did. Totally, 718 patients with PDAC (median size of 2.8 cm) and 661 healthy control cases with more than 40,000 CT images were included and divided into the training, local validation (same-race individuals’ data set), and external validation (different-race individuals’ data set) groups [13]. The eleven most relevant radiomics features were extracted from 88 relevant features for building the classification models [13]. A patch-based model showed a high AUC (0.97), with 92.2% sensitivity, 90.2% specificity, and 90.6% accuracy in classifying patches of PDAC within a pancreas in the local test group [13], which outperformed the diagnostic performance in our test group. When the model was validated in the external test group, an AUC of 0.83 with 48.3% sensitivity, 93.2% specificity, and 86% accuracy was reported [13]. Considering the overall accuracy, this is comparable with our results for ADC images for the test group, but contrary to the mentioned study [13], low specificity and good sensitivity was achieved in our study (Table 9).

Mukherjee and colleagues compared four ML classifiers on 34 radiomics features selected from pre-diagnostic contrast-enhanced CT scans for the early detection of pancreatic carcinoma and found that SVM outperformed RF, a KNN, and XGBoost in diagnostic accuracy [14]. Liao and co-workers tested five different ML classifying methods (KNN, LR, GNB, SVM, and RF) for the discrimination of high-grade from low-grade PDAC by using the radiomics data extracted from multiphasic contrast-enhanced CT examinations [15]. Similarly to our results, the RF model showed the best predictive ability, with an AUC of 0.943 in the training group, followed by the SVM and GNB [15].

Deng and co-authors used clinical and MRI data to construct radiomics models for the differentiation of PDAC from mass-forming chronic pancreatitis (MFCP) [16]. Comparing multiple models developed on T1W, T2W, arterial (A), and portal (P) phase MR images, the AUCs for each of four MR sequences were consistently higher than for clinical models and radiologists’ evaluations [16]. The portal phase post-contrast MR sequence showed the highest discriminative performance in the primary and validation group [16]. In the primary cohort, the AUCs were 0.893, 0.911, 0.958, 0.997, and 0.516 for T1W, T2W, A, and P phase MR images and clinical models, while in the validation cohort, the corresponding AUC values were 0.882, 0.902, 0.920, 0.962, and 0.649, respectively [16]. The radiomics models outperformed the clinical models and radiologists’ evaluations in both cohorts, with statistical significance (all *p* < 0.05) and good model calibration [16]. These findings suggest that radiomics models based on multiparametric MRI data have great potential for non-invasive differentiation of PDAC from MFCP [16].

Podina et al. made a review of recent studies that used convolutional neural networks (CNNs) that showed effectiveness in detecting and segmenting pancreatic tissues and differentiating between benign and malignant lesions [17]. AI models were also used to predict survival, recurrence, and therapy responses in pancreatic cancer patients. All imaging techniques, CT, MRI, and endoscopic ultrasound, were included.

One of the papers by Kaissis et al. from the beforementioned review showed a strong correlation between clinically relevant histopathological subtypes and model predictions, highlighting the potential of quantitative imaging for the pre-operative subtyping and prognosis of PDAC [18].

The use of multiparametric MRI radiomics has been explored by Xie, Fan, and colleagues to predict lymph node metastasis and other survival-related factors in PDAC patients [19]. This approach involves extracting texture features from both peritumoral and intratumoral regions, which were used to train six classifiers. The goal is to improve the accuracy of preoperative evaluations for predicting key pathological characteristics, such as tumor grade, lymph node involvement, and overall survival.

Another key survival-related factor is the development of liver metastasis. A recent study from Yuan et al. demonstrated promising results by combining AI models using MRI radiomics and serological markers to predict liver metastasis in PDAC patients [20].

One study similar to our own proposed a potentially reliable MRI-based radiomics approach for differentiating non-functional neuroendocrine tumors of the pancreas and solid pseudopapillary tumors [21]. These lesions are part of the differential diagnosis for pancreatic adenocarcinoma and their differentiation is of great importance.

A meta-analysis by Lainez Ramos-Bossini et al. confirms that sarcopenia, as determined by CT scans, is an independent predictor of poorer overall and progression-free survival in pancreatic cancer patients [22]. These results could be implemented in a clinical context, combining a detection algorithm such as our own with an automated sarcopenia-detection method. This could serve as both a diagnostic screening and a prognostic tool.

Radiological assessment is crucial for determining the stage and management of PDAC, but due to the disease’s heterogeneity and complex tumor microenvironment, traditional morphological evaluation alone cannot accurately predict its aggressiveness or prognosis. Artificial intelligence (AI) and multiparametric magnetic resonance imaging (mpMRI), using specific contrast agents and techniques, can offer high-quality images that provide both morphological and functional insights, helping quantify intratumor characteristics [23].

We selected two basic MR sequences for the analysis, T2W-FS and ADC. T2W-FS was selected as a short-duration and robust MR sequence in order to minimize the inter-subject variations of fat content in the pancreatic tissue. ADC gives better results than T2W-FS images even in the test phase. Although the whole analysis process was performed based on the subject-wise data division, there are still differences in the results obtained for different image databases. Even though the image databases were collected in the same clinic, using the same MR machine and scanning protocol, there was a time difference between the acquisition of the two datasets. During this time, the machine aged and was recalibrated several times, influencing the image quality and resulting in a discrepancy between the datasets. Furthermore, on average, the size of pancreatic tumor in the test group was smaller than in the training–validation group (2.7 cm vs. 3.4 cm), with a greater proportion of tumors smaller than 2 cm (24% vs. 8%). Nonetheless, we showed that pancreatic adenocarcinoma can be differentiated from surrounding normal pancreatic tissue by using an RF classifier and the previously listed nine features from the co-occurrence matrix category from the T2W-FS and ADC MR images, an approach which can help in everyday clinical practice due to its solid sensitivity rate.

Our study has many limitations. First is the relatively small sample size. Second, we performed 2D segmentation that was restricted to three slices instead of 3D volume segmentation of the whole pancreas and tumor. Third, segmentation was performed by a single radiologist. Fourth, there was a lack of randomization of the test group with the training–validation group in terms of tumor size. Fifth, there was an absence of dynamic post-contrast sequences in the study for the sake of making a shortened protocol for screening purposes. Sixth, patients with Wirsung’s duct dilation with secondary atrophy of normal pancreatic tissues were included by only segmenting the small part of the remaining healthy tissue, usually in the processus uncinatus. Seventh, we did not test the model on patients without pancreatic cancer, which limits its validation. A more comprehensive test would involve comparing patients with and without pancreatic cancer to assess the model’s ability to distinguish between the two. All patients in the study dataset had pancreatic cancer, which may not fully demonstrate the model’s effectiveness in distinguishing it from normal pancreas tissue. Finally, there is the absence of an external test group, which could bring into question the generalizability of the classifiers we found.

Our study includes the CLEAR checklist [24], which is a tool designed to improve the design and reporting of clinical radiomics research (see Appendix A). It provides a structured framework to enhance the quality and standardization of scientific communication, guiding authors and reviewers through key concepts in the field.

Early and accurate diagnosis of pancreatic cancer is crucial for improving patient outcomes, and artificial intelligence (AI), particularly deep learning (DL) and radiomics, offers promising advancements in this area. Few reviews discuss the latest developments in AI for pancreatic cancer diagnosis using cross-sectional imaging techniques, with most focusing on CT [7,8,9,25,26,27]. Recent advances include the application of deep learning techniques like convolutional neural networks (CNNs), transformer-based models, and new architectures that focus on handling various types of pancreatic lesions, segmenting multiple organs and tumors, and integrating auxiliary information [7,8,9,24,25,26].

## 5. Conclusions

This study shows that a machine learning approach using radiomics features extracted from only two MRI sequences, T2W-FS and ADC, achieved relatively high sensitivity in the differentiation of pancreatic adenocarcinoma from healthy pancreatic tissue, which could be especially applicable for screening purposes, considering non-ionizing nature of MRI.

In order to be more precise, it would be necessary to collect larger dataset from different MR scanners, perform segmentation by few radiologists, and then redo the in-depth analysis.

## Figures and Tables

**Figure 1 cancers-17-01119-f001:**
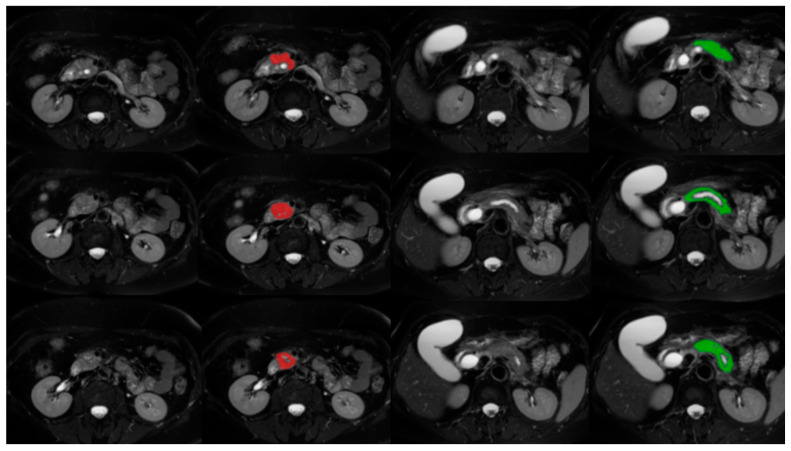
Segmentation of tumor and normal pancreas tissue on T2W-FS images using the MaZda program: adenocarcinoma in the pancreatic head (marked in red) and normal pancreatic tissue (marked in green) on three consecutive slices on the largest tumor area.

**Figure 2 cancers-17-01119-f002:**
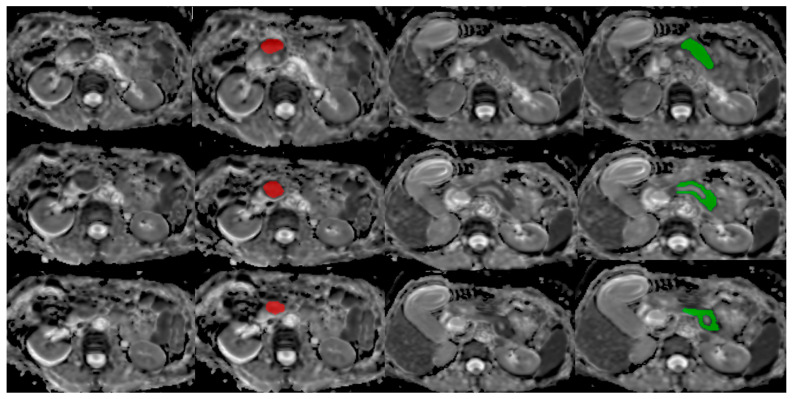
Segmentation of tumor and normal pancreas tissue on ADC maps for the combination of b-values 0 and 800 s/mm^2^ using the MaZda program: adenocarcinoma (marked in red) and normal pancreatic tissue (marked in green) on three consecutive slices.

**Figure 3 cancers-17-01119-f003:**
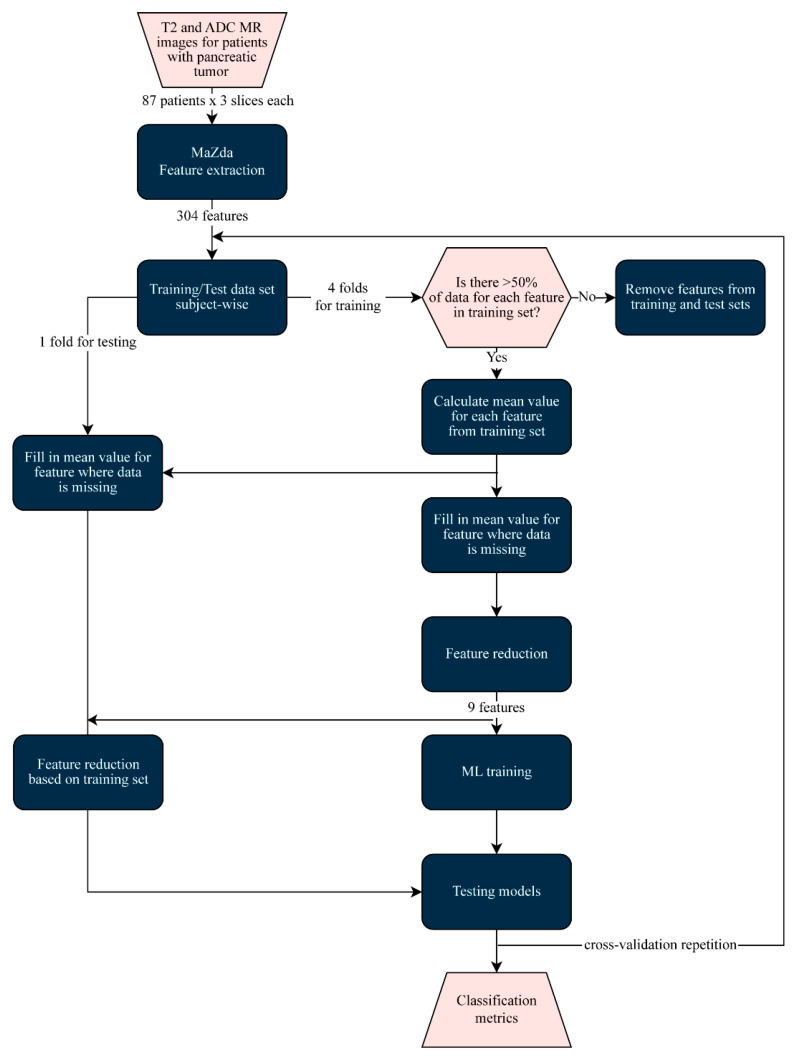
The flowchart of the process of analysis.

**Figure 4 cancers-17-01119-f004:**
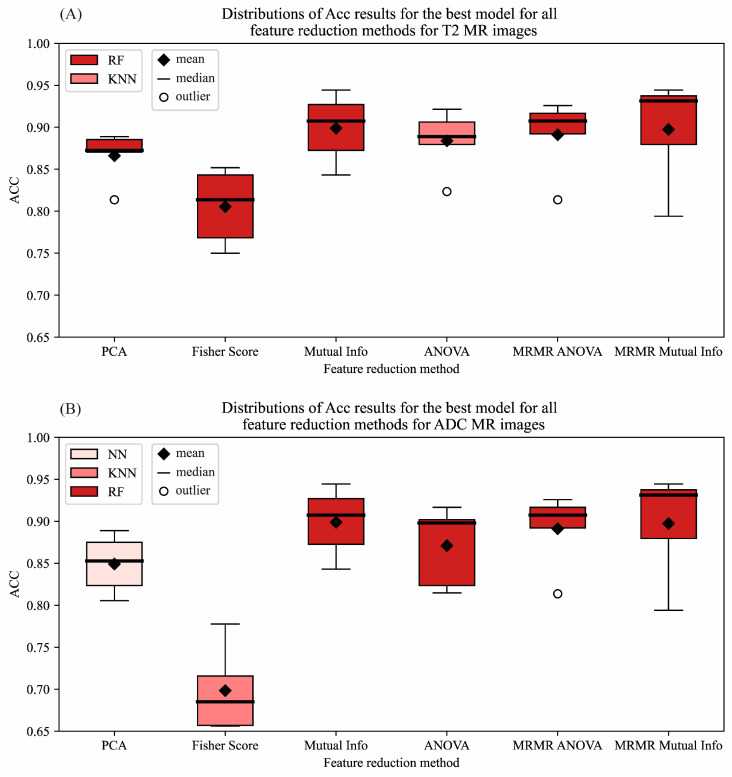
The distributions of Acc results for the best classification model for all feature reduction methods for (**A**) T2W-FS and (**B**) ADC MR images from the first image database.

**Figure 5 cancers-17-01119-f005:**
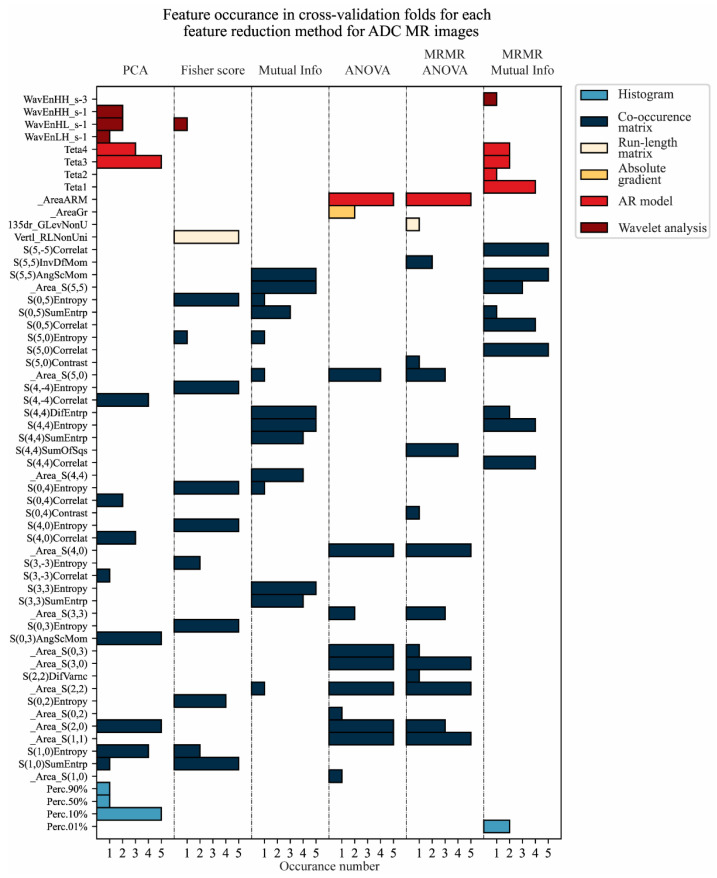
Feature occurrence through cross-validation for each feature reduction method for T2W-FS MR images from the first image database.

**Figure 6 cancers-17-01119-f006:**
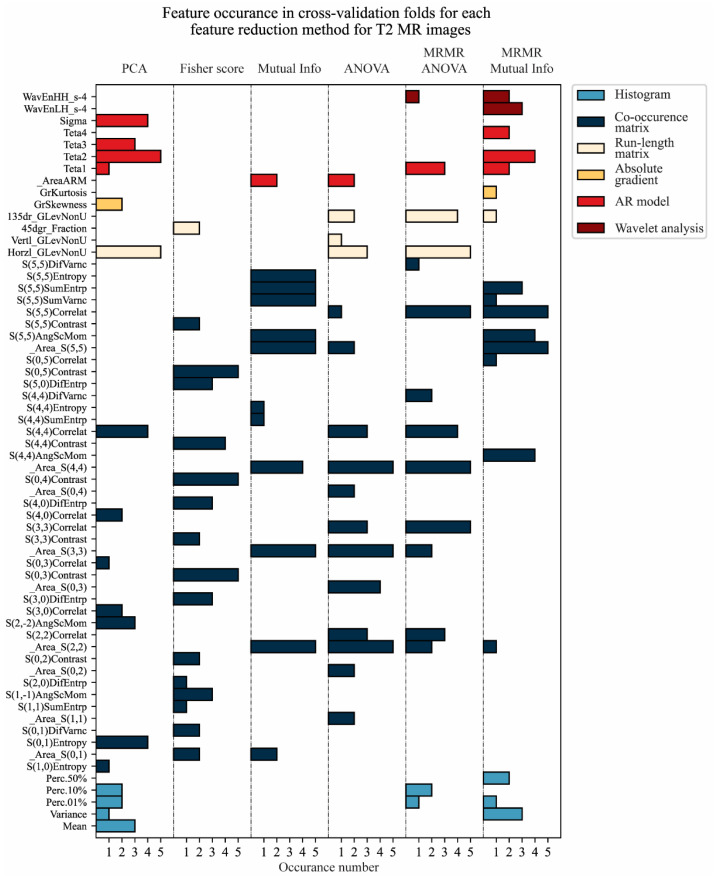
Feature occurrence through cross-validation for each feature reduction method for ADC MR images from the first image database.

**Table 1 cancers-17-01119-t001:** Parameters of MRI scanning protocol.

Parameter	T2-Weighted Single Shot FSE	T2-Weighted FS	T1-Weighted in-Phase	T1-Weighted out of Phase	T1-Weighted 3D-GRE	DW SSSE EPI
TR (ms)	1200	1200	160	160	6.7	5000
TE (ms)	90	90	2	2	4.3	52
Flip angle	90	90	80	80	15	180
BW/pixel (Hz)	62.5	62.5	62.5	62.5	83.33	250
Matrix (phase × frequency)	224 × 288	224 × 288	192 × 256	192 × 256	192 × 320	136 × 136
FOV (mm)	40	40	40	40	40	36
Section thickness (mm)	5	5	5	5	4.4	7
Intersectional gap (%)	20	20	20	20	50	0
No. of signal acquisition	4	1	1	1	1	3
Fat suppression	None	Fat sat	None	None	Fat sat	None
Respiratory control	BH	BH	BH	BH	BH	RT

TR, repetition time; TE, echo time; Hz, Hertz; FOV, field of view; FSE, fast spin echo; GRE, gradient recalled echo; FS, fat suppressed; 3D, three dimensional; DW, diffusion weighted; SSSE, single shot spin-echo; EPI, echo planar imaging; BW, bandwidth; BH, breath hold; RT, respiratory triggered.

**Table 2 cancers-17-01119-t002:** Used classifiers with their observed hyperparameters. The listed parameters are as in the Python libraries and the ones that are not listed are used with their default values.

Logistic Regression	Penalty ∈ {l1, l2} C ∈ {0.1, 0.5, 1, 2} Solver ∈ {newton-cg, lbfgs, liblinear, sag, saga}
k-Nearest Neighbors	N neighbors ∈ {2, 5, 10, 20} Weights ∈ {uniform, distance} Algorithm ∈ {ball tree, kd tree, brute}
Support Vector Machine	C ∈ {0.1, 0.5, 1, 2} Kernel ∈ {linear, poly, rbf, sigmoid}
Neural Network	Hidden layer sizes ∈ {50, 100, 200, [200, 100], [200, 50], [100, 50], [50, 50], [200, 150, 100, 50], [200, 100, 50]} Batch size ∈ {16, 32, 64} Solver ∈ {lbfgs, adam}
Random Forest	N estimators ∈ {200, 400, 600, 800, 1500, 3000} Criterion ∈ {gini, entropy} Max depth ∈ {10, 20, 30, 40, 50, None} Max features ∈ {sqrt, log2}
Gaussian Naïve Bayes	/

**Table 3 cancers-17-01119-t003:** Demographic characteristics of patients and characteristics of pancreatic tumor in the training and validation group.

	Training–Validation Group (*n* = 87)	Test Group (*n* = 58)	*p*
Age (median)	63	63.5	0.913
Gender (Male/Female)	45/42	26/32	0.416
Location of tumor (Head/Body/Tail)	68/15/4	44/12/2	0.836
Diameter of tumor (median)	3.3 cm	2.7 cm	**0.001** **
Diameter of tumor (<2 cm/≥2 cm)	7/80	14/44	**0.009** **

**: highly statistically significant parameter (*p* < 0.01).

**Table 4 cancers-17-01119-t004:** The results for all classifiers with feature reduction based on the Mutual Information score based on the best performance of RF classifier for T2W-FS MR images from the first image database.

	Acc	Se	Sp	F1-Score	AUC
LR	87.04%	85.19%	88.89%	86.79%	87.04%
KNN	89.81%	90.74%	88.89%	89.91%	89.81%
SVM	90.74%	88.89%	92.59%	90.57%	90.74%
RF	94.44%	94.44%	94.44%	94.44%	94.44%
NN	89.81%	85.19%	94.44%	89.32%	89.81%
GNB	75.00%	61.11%	88.89%	70.97%	75.00%

**Table 5 cancers-17-01119-t005:** The results for the RF classifier with all feature reduction methods based on the best performance of the RF classifier for T2W-FS MR images from the first image database.

	Acc	Se	Sp	F1-Score	AUC
PCA	88.89%	90.74%	87.04%	89.09%	88.89%
Fisher Score	76.85%	83.33%	70.37%	78.26%	76.85%
Mutual Information	94.44%	94.44%	94.44%	94.44%	94.44%
ANOVA	89.81%	90.74%	88.89%	89.91%	89.81%
MRMR ANOVA	92.59%	90.74%	94.44%	92.45%	92.59%
MRMR Mutual Information	94.44%	96.30%	92.59%	94.55%	94.44%

**Table 6 cancers-17-01119-t006:** The results for all classifiers with feature reduction based on the Mutual Information score based on the best performance of the RF classifier for ADC MR images from the first image database.

	Acc	Se	Sp	F1-Score	AUC
LR	95.37%	94.44%	96.30%	95.33%	95.37%
KNN	99.07%	100%	98.15%	99.08%	99.07%
SVM	96.30%	96.30%	96.30%	96.30%	96.30%
**RF**	**98.15%**	**98.15%**	**98.15%**	**98.15%**	**98.15%**
NN	91.67%	98.15%	85.19%	92.18%	91.67%
GNB	91.67%	92.59%	90.74%	91.74%	91.67%

**Table 7 cancers-17-01119-t007:** The results for the RF classifier with all feature reduction methods based on the best performance of the RF classifier for ADC MR images from the first image database.

	Acc	Se	Sp	F1-Score	AUC
PCA	88.89%	96.30%	81.48%	89.66%	88.89%
Fisher Score	91.67%	98.15%	85.19%	92.18%	91.67%
**Mutual Information**	**98.15%**	**98.15%**	**98.15%**	**98.15%**	**98.15%**
ANOVA	91.67%	92.59%	90.74%	91.74%	91.67%
MRMR ANOVA	92.59%	94.44%	90.74%	92.72%	92.59%
MRMR Mutual Information	97.22%	98.15%	96.30%	97.25%	97.22%

**Table 8 cancers-17-01119-t008:** The hyperparameters for the best performing RF model and ranked selected features after the feature reduction based on the Mutual Information score for both T2W-FS and ADC MR images from the first image database.

T2W-FS		ADC	
RF hyperparameters	N estimators = 200 Criterion = entropy Max depth = 10 Max features = sqrt	RF hyperparameters	N estimators = 200 Criterion = gini Max depth = 10 Max features = sqrt
Features	_Area_S(0,1) _Area_S(2,2) _Area_S(3,3) _Area_S(4,4) _Area_S(5,5) S(5,5)AngScMom S(5,5)SumVarnc S(5,5)SumEntrp S(5,5)Entropy	Features	S(3,3)SumEntrp S(3,3)Entropy _Area_S(4,4) S(4,4)SumEntrp S(4,4)Entropy S(4,4)DifEntrp S(0,5)SumEntrp _Area_S(5,5) S(5,5)AngScMom

**Table 9 cancers-17-01119-t009:** The results for RF classifier with 9 selected features for second image database.

	Acc	Se	Sp	F1-Score	AUC
T2W-FS	69.25%	86.21%	52.30%	73.71%	69.25%
ADC	81.32%	92.53%	70.10%	83.21%	81.32%

## Data Availability

Data are contained within the article.

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
