# Peer review of "Applicability of Radiomics for Differentiation of Pancreatic Adenocarcinoma from Healthy Tissue of Pancreas by Using Magnetic Resonance Imaging and Machine Learning"

_cancers, 2025, doi:10.3390/cancers17071119_

Round 1
Reviewer 1 Report
Comments and Suggestions for Authors
The Authors present a study that analyzes different classification models to differentiate pancreatic adenocarcinoma from the surrounding healthy pancreatic tissue based on radiomic analysis of magnetic resonance (MR) images. The Authors conclude that the machine learning approach using radiomic features extracted from T2W-FS and ADC achieved relatively high sensitivity in differentiating pancreatic adenocarcinoma from healthy pancreatic tissue, which could be particularly applicable for screening purposes.
The study has limitations reported by the Authors themselves: the small sample size, 2D segmentation limited to three sections, segmentation performed by a single Radiologist, and the lack of randomization between the test group and the validation group. However, the originality of the study lies in the lack of literature using MRI radiomic data for differentiating pancreatic adenocarcinoma from pancreatic tissue.
Since the study involves patients from a hospital database and uses patient data and images, explicit authorization from the Ethics Committee is required. The Authors only report that there is approval from the local Ethics Committee. Greater clarity is needed on this point.
I believe it can be accepted for publication if the Authors resolve the issue related to the Ethics Committee. Minor Revision.
Reviewer 2 Report
Comments and Suggestions for Authors
This paper explores the utility of radiomics from T2 and ADC MRI sequences in differentiating PDAC from normal pancreas. The paper is interesting and provides a well-organized methodological processing (with some improvement areas indicated below). However, the paper needs deep refinement in writing terms and should further justify the clinical utility of their findings, among other suggestions. They are recommended to carry out major changes.
General comments.
Profound English correction of the manuscript is required, as there are many writing errors (e.g., in the title it is incorrect to say “the” before magnetic resonance, absence of “the” before “differentiation”), inhomogeneous use of acronyms (e.g., MRI -magnetic resonance imaging-) and terms (e.g., “pancreatic adenocarcinoma” and “pancreatic carcinoma”), incorrect sentences with lack of subjects or verbs, etc. Profound English writing review is necessary.
Simple summary: should be focused on methodology, main findings and clinical utility.
Introduction
The aims of the study are poorly explained at the end of the introduction section. You should clearly explain what the main purpose of the study is, and what is/are the secondary purpose/s. In my opinion, the justification of the screening utility of radiomics should be better and further explained.
Methods
It is not clear why exactly 3 consecutive slices were obtained for each tumor. This should not be a constraining limitation for analyses, and some tumors may be larger (or even smaller). I see that the authors include this as a limitation of the study, but it should be further clarified with this criterion was initially adopted.
In addition, data imputation should be further justified. Did any previous study use the same criteria followed by the authors? How could this influence the obtained results? This should be added to the discussion.
The authors are recommended to include (and follow) the CLAIM or CLEAR guidelines for a more structured and evidence-based support of their paper:
Tejani AS, Klontzas ME, Gatti AA, Mongan JT, Moy L, Park SH, Kahn CE Jr; CLAIM 2024 Update Panel. Checklist for Artificial Intelligence in Medical Imaging (CLAIM): 2024 Update. Radiol Artif Intell. 2024 Jul;6(4):e240300. doi: 10.1148/ryai.240300. PMID: 38809149; PMCID: PMC11304031.
Kocak B, Baessler B, Bakas S, Cuocolo R, Fedorov A, Maier-Hein L, Mercaldo N, Müller H, Orlhac F, Pinto Dos Santos D, Stanzione A, Ugga L, Zwanenburg A. CheckList for EvaluAtion of Radiomics research (CLEAR): a step-by-step reporting guideline for authors and reviewers endorsed by ESR and EuSoMII. Insights Imaging. 2023 May 4;14(1):75. doi: 10.1186/s13244-023-01415-8. PMID: 37142815; PMCID: PMC10160267.
Results and Discussion
I wonder what happens when a tumor causes Wirsung’s duct dilation with secondary atrophy, as this is not “healthy pancreatic tissue”. How was this potential bias controlled? Otherwise, please indicate such limitation in the discussion of the study.
The introduction and discussion are relatively well organized, but there are few (19) references. Considering the clinical problem addressed (pancreatic cancer, imaging-based assessment methods), you should further discuss several points. For instance, in the introduction, you could emphasize the role of imaging in pancreatic cancer thanks to automated detection and radiomics tools, and in the discussion you should emphasize how your results could be implemented in a clinical context; for instance, combining a detection algorithm with an automated sarcopenia-detection method so as to provide both a diagnostic screening and prognostic tool). Here are some recent suggested papers, but at least 25-30 references should be included:
Zhao B, Cao B, Xia T, Zhu L, Yu Y, Lu C, Tang T, Wang Y, Ju S. Multiparametric MRI for Assessment of the Biological Invasiveness and Prognosis of Pancreatic Ductal Adenocarcinoma in the Era of Artificial Intelligence. J Magn Reson Imaging. 2025 Jan 9. doi: 10.1002/jmri.29708. Epub ahead of print. PMID: 39781607.
Láinez Ramos-Bossini AJ, Gámez Martínez A, Luengo Gómez D, Valverde-López F, Morillo Gil AJ, González Flores E, Salmerón Ruiz Á, Jiménez Gutiérrez PM, Melguizo C, Prados J. Computed Tomography-Based Sarcopenia and Pancreatic Cancer Survival-A Comprehensive Meta-Analysis Exploring the Influence of Definition Criteria, Prevalence, and Treatment Intention. Cancers (Basel). 2025 Feb 11;17(4):607. doi: 10.3390/cancers17040607. PMID: 40002202; PMCID: PMC11853262.
Podină N, Gheorghe EC, Constantin A, Cazacu I, Croitoru V, Gheorghe C, Balaban DV, Jinga M, Țieranu CG, Săftoiu A. Artificial Intelligence in Pancreatic Imaging: A Systematic Review. United European Gastroenterol J. 2025 Feb;13(1):55-77. doi: 10.1002/ueg2.12723. Epub 2025 Jan 26. PMID: 39865461; PMCID: PMC11866320.
Another limitation or area of further research is related to the MRI sequences used. The authors could emphasize the role of other sequences, particularly dynamic post-contrast ones.
The last paragraph of the discussion section seems randomly allocated; the information contained in the said paragraph may be relevant but should be place in an appropriate location in the manuscript.
Finally, the authors are encouraged to further discuss and compare the radiomic features extracted in terms of model explainability, as this is increasingly important in current imaging-AI-related papers.
Comments on the Quality of English Language
Profound English correction of the manuscript is required, as there are many writing errors (e.g., in the title it is incorrect to say “the” before magnetic resonance, absence of “the” before “differentiation”), inhomogeneous use of acronyms (e.g., MRI -magnetic resonance imaging-) and terms (e.g., “pancreatic adenocarcinoma” and “pancreatic carcinoma”), incorrect sentences with lack of subjects or verbs, etc. Profound English writing review is necessary.
Reviewer 3 Report
Comments and Suggestions for Authors
The authors address the important question of using radiomic features to distinguish pancreatic adenocarcinoma from normal pancreatic tissue. They employ a machine learning approach to determine parameters using T2W-FS and ADC MRI that most accurately distinguish pancreatic carcinoma from normal pancreatic tissue. Using 87 patients in the training-validation group and 58 patients in the test group, the authors found T2W-FS and ADC parameters with high AUC for the training group and overall found ADC to be a better predictor than T2W-FS. While posing an interesting research question, the study has significant limitations:
- Why were the specific classifiers for T2W-FS and ADC chosen?
- While the training-validation cohort had impressive AUC values for the classifiers, the results were considerably worse for the testing cohort. These was also no external test group. This brings into question the generalizability of the classifiers that the authors found.
- The authors' determination of normal pancreas tissue vs cancer relied on segmentation by one radiologist. Given the difficulty of distinguishing normal tissue vs. cancer, it would make more sense to have this assessment be a consensus from multiple radiologists.
- It is interesting that the authors did not do any testing of their model on patients without pancreatic cancer. A good validation of whether their model can distinguish pancreatic cancer from normal pancreas would be in patients with and without pancreatic cancer. All patients in the dataset had pancreatic cancer.
The manuscript would benefit from grammatical review as well as proofreading for typos - for instance in section 2.1 of the methods using "hemotherapy" when the authors mean "chemotherapy".
Round 2
Reviewer 2 Report
Comments and Suggestions for Authors
The authors have appropriately addressed my previously expressed concerns, significantly improving the soundness, clarity and completeness of the study. In my opinion, the paper is now suitable for publication
Comments on the Quality of English LanguageEnglish is understandable but needs proofreading to correct several mistakes.
Reviewer 3 Report
Comments and Suggestions for Authors
The authors have added some commentary regarding some a few of the points initially raised. However, without testing a normal pancreas vs pancreatic cancer cohort, this study is still of very limited utility in distinguishing between tumor and normal pancreas.
Comments on the Quality of English LanguageGrammatical errors remain significant in the manuscript.